# Pythae: Unifying Generative Autoencoders in Python
# A Benchmarking Use Case

**Clément Chadebec**
Université Paris Cité, INRIA, Inserm, SU
Centre de Recherche des Cordeliers *
clement.chadebec@inria.fr

**Louis J. Vincent**
Implicity †
Université Paris Cité, INRIA, Inserm, SU
Centre de Recherche des Cordeliers *
louis.vincent@inria.fr

**Stéphanie Allassonnière**
Université Paris Cité, INRIA, Inserm, SU
Centre de Recherche des Cordeliers *
stephanie.allassonniere@inria.fr

## Abstract

In recent years, deep generative models have attracted increasing interest due to their capacity to model complex distributions. Among those models, variational autoencoders have gained popularity as they have proven both to be computationally efficient and yield impressive results in multiple fields. Following this breakthrough, extensive research has been done in order to improve the original publication, resulting in a variety of different VAE models in response to different tasks. In this paper we present **Pythae**, a versatile *open-source* Python library providing both a *unified implementation* and a dedicated framework allowing *straightforward*, *reproducible* and *reliable* use of generative autoencoder models. As an example of application, we propose to use this library to perform a case study benchmark where we present and compare 19 generative autoencoder models representative of some of the main improvements on downstream tasks such as image reconstruction, generation, classification, clustering and interpolation. The open-source library can be found at `https://github.com/clementchadebec/benchmark_VAE`.

## 1 Introduction

Over the past few years, generative models have proven to be a promising approach for modelling datasets with complex inherent distributions such a natural images. Among those, Variational AutoEncoders (VAE) [35, 54] have gained popularity due to their computational efficiency and scalability, leading to many applications such as speech modelling [10], clustering [24, 66], data augmentation [14] or image generation [52]. Similarly to autoencoders, these models encourage good reconstruction of an observed input data from a latent representation, but they further assume latent vectors to be random variables involved in the generation process of the observed data. This imposes a latent structure wherein latent variables are driven to follow a prior distribution that can then be used to generate new data. Since this breakthrough, various contributions have been made to enrich the original VAE scheme through new generating strategies [24, 63, 27, 5, 48], reconstruction objectives [39, 59] and more adapted latent representations [29, 34, 64, 3, 14] to cite a few. A drawback of VAEs is that due to the intractability of the log-likelihood objective function, VAEs have to resort

---

* 15 Rue de l'École de Médecine, 75006 Paris
† `https://www.implicity.com` - Implicity Paris, France.

36th Conference on Neural Information Processing Systems (NeurIPS 2022) Track on Datasets and Benchmarks.

to optimizing a lower bound on the true objective as a proxy, which has been mentionned as a major limitation of the model [11, 1, 29, 21, 67]. Hence, extensive research has been proposed to improve this bound through richer distributions [55, 53, 36, 13]. More recently, it has been shown that autoencoders can be turned into generative approaches through latent density estimation [27], extending the concept of *Generative AutoEncoders* (GAE) to a more general class of autoencoder models.

Nonetheless, most of this research has been done in parallel across disjoint sub-fields of research and to the best of our knowledge little to no work has been done on homogenising and integrating these distinct methods in a common framework. Moreover, for many of the aforementioned publications, implementations may not be available or maintained, therefore requiring time-consuming re-implementation. This induces a strong bottleneck for research to move forward in this field and makes reproducibility challenging, which calls for the need of a unified generative autoencoder framework. To address this issue we introduce **Pythae** (**Pyth**on **A**uto**E**ncoder), a versatile open source Python library for generative autoencoders providing unified implementations of common methods, along with a reproducible framework allowing for easy model training, data generation and experiment tracking. We then propose to illustrate the usefulness of the proposed library on a benchmark case study of 19 generative autoencoder methods on classical image datasets. We consider five different downstream tasks: image reconstruction and generation, latent vector classification and clustering, and image interpolation on three well known imaging datasets.

## 2 Variational autoencoders

In this section, we recall the original VAE setting and present some of the main improvements that were proposed to enhance the model.

### 2.1 Background

Given $x \in \mathbb{R}^D$, a set of observed variables deriving from an unknown distribution $p(x)$, a VAE assumes that there exists $z \in \mathbb{R}^d$ such that $z$ is a latent representation of $x$. The generation process of $x$ thus decomposes as

$$p_\theta(x) = \int_{\mathcal{Z}} p_\theta(x|z) p_z(z) dz \,, \tag{1}$$

where $p_z$ is the *prior distribution* on the latent space $\mathbb{R}^d$. The distribution $p_\theta(x|z)$ is referred to as the *decoder* and is modelled with a simple parametric distribution whose parameters are given by a neural network. Since the true posterior $p_\theta(z|x)$ is most of the time intractable due to the integral in Eq. (1) recourse to Variational Inference [31] is needed and a variational distribution $q_\phi(z|x)$ which we refer to as the *encoder* is introduced. The approximate posterior $q_\phi$ is again taken as a simple parametric distribution whose parameters are also modelled by a neural network. This allows to define an unbiased estimate $\widehat{p}_\theta$ of the marginal distribution $p_\theta(x)$ using importance sampling with $q_\phi(z|x)$ *i.e.* $\widehat{p}_\theta(x) = \frac{p_\theta(x|z) p_z(z)}{q_\phi(z|x)}$ and $\mathbb{E}_{z \sim q_\phi}[\widehat{p}_\theta] = p_\theta$. Applying Jensen's inequality leads to a lower bound on the likelihood given in Eq. (1):

$$\log \underbrace{\mathbb{E}_{z \sim q_\phi}[\widehat{p}_\theta(x)]}_{p_\theta(x)} \geq \mathbb{E}_{z \sim q_\phi}[\log \widehat{p}_\theta(x)] = \underbrace{\mathbb{E}_{z \sim q_\phi}[\log p_\theta(x|z)]}_{\text{reconstruction}} - \underbrace{\mathcal{D}_{KL}[q_\phi(z|x)||p_z(z)]}_{\text{regularisation}} \,, \tag{2}$$

where $\mathcal{D}_{KL}(p||q)$ is the Kullback-Leibler divergence between distributions $p$ and $q$. This bound is referred to as the Evidence Lower Bound (ELBO) [35] and is used as the training objective to maximize in the traditional VAE scheme. It can be interpreted as a two terms objective [27] where the *reconstruction* loss forces the output of the decoder to be close to the original input $x$, while the *regularisation* loss forces the posterior distribution $q_\phi(z|x)$ outputted by the encoder to be close to the prior distribution $p_z(z)$. Under standard VAE assumption, the prior distribution is a multivariate standard Gaussian $p_z = \mathcal{N}(0, I_d)$, the approximate posterior is set to $q_\phi(z|x) = \mathcal{N}(z|\mu(x), \Sigma(x))$ where $(\mu(x), \Sigma(x))$ are outputs of the encoder network.

### 2.2 Improvements upon the classical VAE method

Building on the breakthrough of VAEs, several papers have proposed improvements to the model. In this section we present 4 axes which we consider to be representative of the major advancements

made on VAEs, as well as classical models characterising the main improvements within each of these axes.

**Improving the prior**  It has been shown that the role of the prior distribution $p_z$ is crucial in the good performance of the VAE [30] and choosing a family of overly simplistic priors can lead to over-regularization [20] and poor reconstruction performance [22]. In particular, it was shown that the prior maximizing the ELBO objective is the *aggregated posterior* $q(z) = \frac{1}{N} \sum_{i=1}^{N} q_\phi(z|x_i)$ [63]. However, it should be noted that a perfect fit between the prior and the aggregated posterior is not necessarily desired since it has been shown in [6, 63] that it may lead to over-fitting as it essentially amounts to the model memorising the training set. Hence, multi-modal priors [46, 24, 63] were proposed, followed by hierarchical latent variable models [60, 37] and prior learning based approaches [17, 2] to address the poor expressiveness of the prior distribution and model richer generative distributions. Considering a specific geometry of the latent space also led to alternative priors taking into account geometrical aspects of the latent space [23, 25, 43, 3, 15, 57, 33, 14]. Another interesting approach proposed for instance in [64, 27] consists in using density estimation post training with another distribution or normalising flows [53] on the learned latent codes.

**Towards a better lower bound**  Another major axis of improvement of the VAE model has been to tighten the gap between the ELBO objective and the true log probability [11, 1, 29, 21, 67]. The ELBO objective can indeed be written as the difference between the true log probability and a KL divergence between the approximate posterior and the true posterior

$$\mathcal{L}_{\text{ELBO}}(x) = \log p_\theta(x) - \mathcal{D}_{KL}\big[q_\phi(z|x)||p(z|x)\big] . \tag{3}$$

Hence, if one wants to make the ELBO gap tighter, particular attention should be paid to the choice in the approximate posterior $q_\phi(z|x)$. In the original model, $q_\phi(z|x)$ is chosen as a simple distribution for tractability of the ELBO in Eq. (2). However, several approaches have been proposed to extend the choice of $q_\phi$ to a wider class of distributions using MCMC sampling [55] or normalising flows [53]. For instance, Kingma et al. [36] improve upon the works of [53] with an inverse auto-regressive normalising flow (IAF), a new type of normalizing flow that better scales to high-dimensional latent spaces. With this objective in mind a Hamiltonian VAE aimed at targeting the true posterior during training with a Hamiltonian Monte Carlo [47] inspired scheme was proposed [13] and extended to Riemannian latent spaces in [14].

**Encouraging disentanglement**  Although there is no clear consensus upon the definition of disentanglement, it is commonly referred to as the independence between features in a representation [44]. This is a desirable behaviour for VAEs, as it is argued that disentangled features may be more representative and interpretable [29]. In that regard, several approaches have been proposed encouraging a better disentanglement of the features in the latent space. Higgins et al. [29] first argue that increasing the weight of the KL divergence term in the ELBO loss enforces a higher disentanglement of the latent features as the posterior probability is forced to match a multivariate normal standard Gaussian. Following this idea, [12] propose to achieve disentanglement by gradually increasing the proximity between the posterior and the prior [12]. Other methods challenge the view that disentanglement can be achieved by simply forcing the posterior to match the prior, or raise the point that in this case disentanglement is achieved at the cost of a bad reconstruction. From these observations, new approaches arise such as [34] who augment the VAE objective with a penalty that encourages factorial representation of the marginal distributions, or [16] that enforce a penalty on the total correlation favouring disentanglement.

**Amending the distance between distributions**  It can be stressed that the reconstruction term $\mathbb{E}_{z \sim q_\phi(z|x)}[\log p_\theta(x|z)]$ in eq. (2) has a crucial role in the reconstruction and that its choice should be dependent of the application. For instance, methods using a discriminator [39] or using a deterministic differentiable loss function [59] acting as a distance between the input data and its reconstruction were also proposed. The second term in the ELBO measures the distance between the approximate posterior and the prior distribution through the KL divergence and it has however been argued that other distances between probability distributions could be used instead. Hence, approaches using a GAN to distinguish samples from the posterior from samples from the prior distribution [42] or methods based on optimal transport have also been proposed [62, 68].

## 3 The Pythae library

**Why Pythae ?**   To the best of our knowledge, although some well referenced libraries grouping different Variational Auto-Encoder methods exist (*e.g.* [61]), there exists no framework providing both adaptable and easy-to-use unified implementations of state-of-the-art Generative AutoEncoder (GAE) methods. This induces both a strong brake for reproducible research and democratisation of the models since implementations might be difficult to adapt to other use-cases, no longer maintained, or completely unavailable.

**Project vision**   Starting from this observation, we created **Pythae**, an open-source python library inspired from [51, 65] providing unified implementations of generative autoencoding methods, allowing for easy use and training of GAE models. Pythae is designed with the following points in mind:

- **Usable by all** Pythae makes GAE models accessible to all - beginners to experts. This means beginners can run *ready-to-use* models with a few lines of code, while more advanced users can easily access and adapt different methods to their specific use-cases, with custom encoder/decoder definition. Indeed, the library was designed to be flexible enough to allow users to use existing implementations on their own data, with custom model hyperparameters, training configurations and network architectures.

    The library has an online documentation[3] and is also explained and illustrated through tutorials available either on a local machine or on the *Google Colab* platform [9].

- **Unified implementation** The brick-like structure of Pythae allows for seamless but efficient interchange between models, sampling techniques, network architectures, model hyperparameters and training schemes. Pythae is unit-tested ensuring code quality and continuous development with a code coverage of 98% as of release 0.6. The library is made available on *pip* and *conda* allowing an easy integration. Its development is performed through releases that ensure stable and robust implementations.

- **A reproducible research environment** Pythae is open to all and as such encourages transparent and reproducible research, as illustrated in the next section. With a variety of different interchangeable models gathered in a common library, it can be used as a sandbox for research and applications. Moreover, the library also integrates an easy-to-use experiment tracking tool (*wandb*) [8] allowing to monitor runs launched with Pythae and compare them through a graphic interface, and an online model sharing tool, the HuggingFace Hub, allowing to share models with peers.

- **Evolving and driven by the community** Pythae's design is intended to evolve with the addition of new models to enrich the existing model base. Furthermore, peers can contribute by reviewing and submitting models to enrich the library, a few of which have already been added at the time of this publication.

**Code structure**   Pythae was thought for easy model training and data generation, while striving for simplicity with a quick and user-friendly model selection and configuration. The backbone of the library is the module *pythae.models* in which all the autoencoder models are implemented. Each model implementation is accompanied with a configuration *dataclass* containing any hyperparameters relative to the model and allowing easing configuration loading and saving from *json* files.

All the models are implemented using a common API allowing for a seamless integration with *pythae.trainers* (for training) and *pythae.samplers* (for generation) along with a simplified usage as illustrated in Fig. 1. In particular, Pythae provides pipelines allowing to train an autoencoder model or to generate new data with only a few lines of code, as shown in Appendix. A.

It mainly relies on the Pytorch [50] framework and in its basic usage only essential hyper-parameter configurations and data (arrays or tensors) are needed to launch a model training or generation. More advanced options allowing further flexibility such as defining custom encoder and decoder neural-networks are also available and can be found in the documentation and tutorials. It can adapt to various types of data through the use of different already-implemented or user provided encoder

---

[3]The full documentation can be found at `https://pythae.readthedocs.io/en/latest/`.

and decoder neural-network architectures. In addition, Pythae also provides several ways to generate new data through different popular sampling methods in the *pythae.samplers* module. We detail some aspects of the library in Appendix. A.

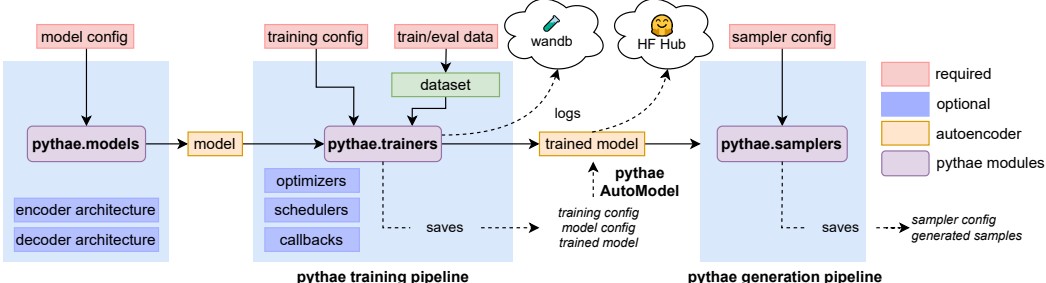

Figure 1: Pythae library diagram

## 4   Case study benchmark

By nature of its structured framework, Pythae allows for easy comparison between models on any chosen task. As an illustrative purpose, we propose a case study where we use Pythae to perform a straightforward benchmark comparison of models implemented in Pythae on a selection of well-known elementary tasks. The aim of these tasks is to underline general trends within groups of GAEs, based on common behaviours, as well as judge the versatility of the models. However, this benchmark should not be considered as a means to rank models on these tasks, as performances depend on sometimes complex hyper-parameter tuning and training, which we consider to be outside of the scope of this benchmarking use case. The scripts used for the benchmark are provided in supplementary materials.

### 4.1   Benchmark setting

In this section, we present the setting of the benchmark. Comprehensive results for all the experiments are available through the monitoring tool [8] used in Pythae to allow complete transparency.

**The data**    To perform the different tasks presented in this paper, 3 classical and widely used image datasets are considered: MNIST [40], CIFAR10 [38] and CELEBA [41]. These datasets are publicly available, widely used for generative model related papers and have well known associated metrics in the literature. Each dataset is split into a train set, a validation set and a test set. For MNIST and CIFAR10 the validation set is composed of the last 10k images extracted from the official train set and the test set corresponds to the official one. For CELEBA, we use the official train/val/test split.

**The models**    We propose to compare 19 generative autoencoder models representative of the improvements proposed in the literature and presented in Sec. 2.2. Descriptions and explanations of each implemented model can be found in Appendix. D. We use as baseline an Autoencoder (**AE**) and a Variational Autoencoder (**VAE**). To assess the influence of a *more expressive prior*, we propose using a VAE with VAMP prior (**VAMP**) [63] and regularised autoencoders with either a gradient penalty (**RAE-GP**) or a L2 penalty on the weights of the decoder (**RAE-L2**) that use *ex-post* density estimation [27]. To represent models trying to reach a *better lower bound*, we choose a Importance Weighted Autoencoder (**IWAE**) [11] and VAEs adding either simple linear normalising flows (**VAE-lin-NF**) [53] or using IAF (**VAE-IAF**) [36]. For *disentanglement-based models*, we select a $\beta$-**VAE** [29], a **FactorVAE** [34] and a $\beta$-**TC VAE** [16]. To stress the *influence of the distance* used between distributions we add a Wasserstein Autoencoder (**WAE**)[62] and an **InfoVAE** [68] with either Inverse Multi-Quadratic (IMQ) or a Radial Basis Function kernel (RBF) together with an Adversarial Autoencoder (**AAE**)[42], a **VAEGAN**[39] and a VAE using structural similarity metric for reconstruction (**MSSSIM-VAE**) [59]. Finally, we add a **VQVAE** [64] since having a discrete latent space has shown to yield promising results. Models implemented in Pythae requiring too much training time or more intricate hyper-parameter tuning were excluded from the benchmarks.

In the following, we will distinguish *AE-based* (autoencoder-based) methods (AE, RAE, WAE and VQVAE) from the other *variational-based* methods.

**Training paradigm** Each of the aforementioned models is equipped with the same neural network architecture for both the encoder and decoder leading to a comparable number of parameters [4]. For each task, 10 different configurations are considered for each model, allowing a simple exploration of the models' hyper-parameters, leading to 10 trained models for each dataset and each neural network type (ConvNet or ResNet) leading to a total of 1140 models[5]. It is important to note that the hyper-parameter exploration is not exhaustive and models sensitive to hyper-parameter tuning may have better performances with a more extensive parameter search. The sets of hyper-parameters explored are detailed in Appendix. D for each model.

## 4.2 Experiments

In this section, we present the main results observed on 5 downstream tasks.

### 4.2.1 Fixed latent dimension

In this first part, latent dimensions are set to 16, 256 and 64 for the MNIST, CIFAR10 and CELEBA datasets respectively, as we observed those latent dimensions to lead to good performances. See results per model and across the 10 configurations specified in Appendix. D to assess the influence of the parameters on the tasks.

**Task 1: Image reconstruction** For each model, reconstruction error is evaluated by selecting the configuration minimising the Mean Square Error (MSE) between the input and the output of the model on the validation set, while results are shown on the test set[6]. We show in Table. 1 the MSE and Frechet Inception Distance[7] (FID) [28] of the reconstructions from this model on the test set. It is important to note that using the MSE as a metric places models using different reconstruction losses (VAEGAN and MSSSIM-VAE) at a disadvantage.

As expected, the autoencoder-based models seem to perform best for the reconstruction task. Nonetheless, this experiment also shows the interest of adding regularisation to the autoencoder since improvements over the AE (RAE-GP, RAE-L2) achieve better performance than the regular AE. Moreover, $\beta$-VAE type models demonstrate their versatility as small enough $\beta$ values can lead to less regularisation, therefore favouring a better reconstruction.

**Task 2: Image generation** We consider an image generation task with the trained models. In this experiment, we also explore different ways of sampling new data, either 1) using a simple distribution chosen as $\mathcal{N}(0, I_d)$ and corresponding to the standard prior for variational approaches ($\mathcal{N}$); 2) fitting a 10 components mixture of Gaussian in the latent space post training as proposed in [27] (GMM), 3) fitting a normalising flow taken as a Masked Autoregressive Flow (MAF) [49] or 4) fitting a VAE in a similar fashion as [22]. For the MAF, two-layer MADE [26] are used. For each sampler, we select the models achieving the lowest FID on the validation set and compute the Inception Score[8] (IS) [56] and the FID on the test set[9]. It should be noted that although the use of IS and FID has been criticised [4, 58, 18, 45, 32], we still choose to use those metrics for clarity's sake as they are within the most commonly used metrics for image generation on generative models. The main results are shown in Table. 3 for the normal and GMM sampler (see Appendix. D for the other sampling schemes).

One of the key findings of this experiment is that performing *ex-post* density (therefore not using the standard Gaussian prior) for the variational approach tends to almost always lead to better generation

---

[4]Some models may actually have additional parameters in their intrinsic structure *e.g.* a VQVAE learns a dictionary of embeddings, a VAMP learns the pseudo-inputs, a VAE-IAF learns auto-regressive flows. Nonetheless, since we work on images, the number of parameters remains in the same order of magnitude.

[5]The training setting (curves, configs ...) can be found at `https://wandb.ai/benchmark_team/trainings` while detailed experimental set-up is available in Appendix. C.

[6]See the whole results at `https://wandb.ai/benchmark_team/reconstructions`

[7]We used the implementation of `https://github.com/bioinf-jku/TTUR`

[8]We used the implementation of `https://github.com/openai/improved-gan`

[9]See the whole results at `https://wandb.ai/benchmark_team/generations`

metrics even when a simple 10-components mixture of Gaussian is used. Interestingly, we note that when a more advanced density estimation model such as a MAF is used, results appear equivalent to those of the GMM (see Appendix. D). This may be due to the simplicity of the database we used and in consequence of the distribution of the latent codes that can be approximated well enough with a GMM. It should nonetheless be noted that the number of components in the GMM remains a key parameter which was set to the number of classes for MNIST and CIFAR10 since it is known, however too high a value may lead to overfitting while a low one may lead to worse results.

**Task 3: Classification**   To measure the meaningfulness of the learned latent representations we perform a simple classification task with a single layer classifier as proposed in [19]. The rationale behind this is that if a GAE succeeds in learning a disentangled latent representation a simple linear classifier should perform well [7]. A single layer classifier is trained in a supervised manner on the latent embeddings of MNIST and CIFAR10. The train/val/test split used is the same as for the autoencoder training. For each model configuration, we perform 20 runs of the classifier on the latent embeddings and define the best hyper-parameter configuration as the one achieving the highest mean accuracy on these 20 runs on the validation set. We report the mean accuracy on the test set across the 20 runs for the selected configuration in Table. 2 (left)[10].

As expected, models explicitly encouraging disentanglement in the latent space such as the $\beta$-VAE and $\beta$-TC VAE achieve better classification when compared to a standard VAE. Nonetheless, AE-based models seem again the best suited for such a task since variational approaches tend to enforce a continuous space, consequently bringing latent representations of different classes closer to each other. As a general observation, we can state that models with a more flexible prior achieve better results on this task.

**Task 4: Clustering**   As a complement to the previous task, performing clustering directly in the latent space of the trained autoencoders can give insights on the quality of the latent representation. Indeed, a well defined latent space will maintain the separation of the classes inherent to the datasets, leading to easy and stable $k$-means performances. To do so, we propose to fit 100 separate runs of the $k$-means algorithm and we show the mean accuracy obtained on the train embeddings in Table. 2 (right)[11]. This experiment allows us to explore and measure the *clusterability* of the generated latent spaces [7]. To measure accuracy we assign the label of the most prevalent class to each cluster.

The conclusions of this experiment are slightly different from the previous one since models targeting disentanglement seem to be equalled by the original VAE. Interestingly, adversarial approaches and other alternatives to the standard VAE KL regularisation method seem to achieve the best results.

**Task 5: Interpolation**   Finally, we propose to assess the ability of the model to perform meaningful interpolations. For this task, we consider a starting and ending image in the test set of MNIST and CIFAR10 and perform a linear interpolation in the generated latent spaces between the two encoded images. We show in Appendix. B the decoding along the interpolation curves. For this task, no metric was found relevant since the notion of "good" interpolation can be disputable. Nonetheless, the obtained interpolated images can be reconstructed and qualitatively evaluated.

For this task, variational approaches were found to obtain better results as the inherent structure the posterior distribution imposes in the latent space results in a "smoother" transition from one image to another when compared to autoencoders that mainly superpose images, especially in higher dimensional latent spaces.

### 4.2.2   Varying latent dimension

An important parameter of autoencoder models which is too often neglected in the literature is the dimension of the latent space. We now propose to keep the same configurations as previously but re-evaluate Tasks 1 to 5 with the latent space varying in the range $[16, 32, 64, 128, 256, 512]$. Results are shown in Fig. 2 for MNIST and a ConvNet[12] (see Appendix. D for CIFAR, ResNet and interpolations). For the generation task, we select the sampler with lowest FID on the validation set.

---

[10]See the whole results at `https://wandb.ai/benchmark_team/classifications`

[11]See the whole results at `https://wandb.ai/benchmark_team/clustering`

[12]MSSSIM-VAE was removed from this plot for visualisation purposes.

**Assessing the influence of the latent dimension**  A clear difference in behaviour is exhibited between variational-based and AE-based methods. For each given task, AEs share a common trend with respect to the evolution of the latent dimension: a common optimal latent dimension within the range $[16, 32, 64, 128, 256, 512]$ is found for each task, but differs drastically among different tasks (*e.g.* 512 for reconstruction, 16 for generation, either 16 or 512 for classification and 512 for clustering with the MNIST dataset). This suggests the existence of a common intra-group optimal latent space dimension for a given task. In addition, we observe that $\beta$-VAE type methods (with the right hyper-parameter choice) can exhibit similar behaviours to AE models. The same observation can be made for variational-based methods, where it is interesting to note that although lower performances are achieved, the apparent optimal latent dimension varies less with respect to the choice of the task. Therefore, a latent dimension of 16 to 32 appears to be the optimal choice for all 4 Tasks on the MNIST dataset, and 32 to 128 on the CIFAR10 dataset. It should be noted that unsupervised tasks such as clustering of the latent representation of the CIFAR10 dataset are hard and models are expected to perform poorly, leading to less interpretable results.

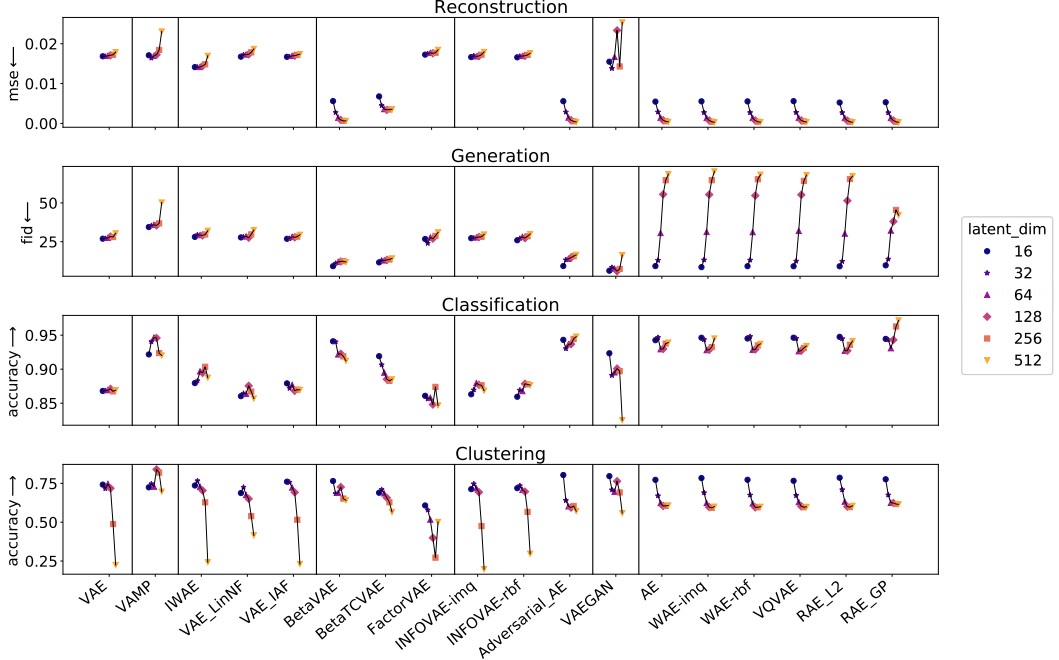

Figure 2: *From top to bottom:* Evolution of the reconstruction MSE, generation FID, classification accuracy and clustering accuracy with respect to the latent space dimension on the MNIST dataset.

## 5    Conclusion

In this paper, we introduce **Pythae**, a new open-source Python library unifying common and state-of-the-art Generative AutoEncoder (GAE) implementations, allowing reliable and reproducible model training, data generation and experiment tracking. This library was designed as an open model testing environment driven by the community, wherein peers are encouraged to contribute by adding their own models, and by doing so favour reproducible research and accessibility to ready-to-use GAE models. As an illustration of the capabilities of Pythae, we perform a benchmarking of 19 generative autoencoder models on 5 downstream tasks (image reconstruction, generation, classification, clustering and interpolation) leading to some interesting findings on the general behaviours of generative autoencoder models. We hope that the library will continue to be adopted by the community and expand thanks to the increasing number of contributions.

Table 1: Mean Squared Error ($10^{-3}$) and FID (lower is better) computed with 10k samples on the test set. For each model, the best configuration is the one achieving the lowest MSE on the validation set.

| Model | ConvNet MNIST (16) MSE ↓ | FID ↓ | CIFAR10 (256) MSE ↓ | FID ↓ | CELEBA (64) MSE ↓ | FID ↓ | ResNet MNIST (16) MSE ↓ | FID ↓ | CIFAR10 (256) MSE ↓ | FID ↓ | CELEBA (64) MSE ↓ | FID ↓ |
|---|---|---|---|---|---|---|---|---|---|---|---|---|
| VAE | 16.85 | 30.71 | 16.24 | 218.66 | 9.83 | 49.22 | 17.24 | 36.06 | 16.33 | 176.63 | 10.59 | 58.75 |
| VAMP | 24.17 | 44.95 | 17.45 | 221.40 | 10.81 | 51.64 | 17.11 | 37.58 | 16.87 | 177.03 | 11.50 | 60.89 |
| IWAE | 14.14 | 34.28 | 16.19 | 237.14 | 9.47 | 50.00 | 15.79 | 38.74 | 16.02 | 183.37 | 10.14 | 60.18 |
| VAE-lin-NF | 16.75 | 31.14 | 16.57 | 221.39 | 9.90 | 49.84 | 17.23 | 36.74 | 16.59 | 177.08 | 10.68 | 58.73 |
| VAE-IAF | 16.71 | 30.64 | 16.33 | 223.65 | 9.87 | 50.05 | 17.05 | 35.98 | 16.39 | 177.05 | 10.63 | 58.41 |
| $\beta$-VAE | 5.61 | 10.55 | 3.60 | 50.55 | 7.28 | **46.96** | 5.87 | 15.81 | 2.40 | 55.67 | 7.78 | 51.59 |
| $\beta$-TC VAE | 6.78 | 14.11 | 5.06 | 53.49 | 7.65 | 50.82 | 7.12 | 18.44 | 4.05 | 66.89 | 8.08 | 52.70 |
| Factor VAE | 17.27 | 30.39 | 16.41 | 224.3 | 10.16 | 53.61 | 18.13 | 37.97 | 16.55 | 176.8 | 10.93 | 59.46 |
| InfoVAE-IMQ | 16.65 | 30.62 | 16.19 | 216.44 | 9.81 | 50.51 | 17.17 | 37.33 | 16.32 | 173.79 | 10.63 | 58.04 |
| InfoVAE-RBF | 16.59 | 30.63 | 16.23 | 217.52 | 9.85 | 50.14 | 17.01 | 37.04 | 16.32 | 175.37 | 10.64 | 58.68 |
| AAE | 5.59 | 10.87 | 2.60 | 40.66 | 7.25 | 50.22 | 5.98 | 17.01 | **2.33** | 55.93 | 7.76 | 50.97 |
| MSSSIM-VAE | 32.60 | 37.91 | 39.42 | 276.70 | 35.60 | 124.52 | 33.67 | 40.25 | 39.61 | 254.34 | 35.43 | 119.92 |
| VAEGAN | 15.49 | **5.54** | 31.40 | 289.35 | 8.91 | 86.58 | 23.25 | **11.35** | 30.22 | 300.07 | 9.32 | 86.32 |
| AE | 5.47 | 11.61 | 2.82 | 41.98 | 7.03 | 51.08 | 6.13 | 13.74 | 2.34 | 55.43 | 7.74 | 50.54 |
| WAE-IMQ | 5.55 | 11.29 | 2.81 | 41.79 | 7.04 | 52.11 | 5.78 | 16.21 | 2.34 | 56.55 | 7.74 | 50.50 |
| WAE-RBF | 5.53 | 11.34 | 2.82 | 42.21 | 7.03 | 51.43 | 5.80 | 16.14 | 2.34 | 56.00 | 7.74 | 51.38 |
| VQVAE | 5.59 | 11.02 | 2.84 | 44.60 | 7.06 | 52.27 | 6.00 | 15.27 | 2.34 | **55.84** | **7.73** | **50.29** |
| RAE-L2 | **5.24** | 15.37 | **2.25** | 49.28 | **6.90** | 53.98 | **5.76** | 17.27 | 2.35 | 57.85 | 7.74 | 51.07 |
| RAE-GP | 5.31 | 12.08 | 2.81 | **41.15** | 7.06 | 51.85 | 5.83 | 15.69 | 2.34 | 56.71 | 7.76 | 51.36 |

Table 2: *Left:* Mean test accuracy of a single layer classifier on the embedding obtained in the latent spaces of each model average on 20 runs. *Right:* Mean accuracy of 100 $k$-means fitted on the training embeddings coming from the autoencoders.

| Model | Classification ConvNet MNIST | CIFAR10 | ResNet MNIST | CIFAR10 | Clustering ConvNet MNIST | CIFAR10 | ResNet MNIST | CIFAR10 |
|---|---|---|---|---|---|---|---|---|
| VAE | 86.75 (0.05) | 32.61 (0.03) | 86.80 (0.03) | 32.37 (0.03) | 69.71 (2.01) | 17.18 (0.68) | 74.21 (0.97) | 18.12 (0.74) |
| VAMP | 92.17 (0.02) | 33.46 (0.17) | 92.58 (0.04) | 33.03 (0.22) | 67.26 (1.25) | 24.03 (0.24) | 72.48 (0.96) | 23.35 (0.11) |
| IWAE | 87.96 (0.04) | 31.86 (0.04) | 88.18 (0.03) | 32.26 (0.04) | 63.93 (1.73) | 19.55 (0.67) | 73.66 (2.30) | 18.44 (0.86) |
| VAE-lin-NF | 86.04 (0.04) | 31.57 (0.02) | 85.85 (0.05) | 31.74 (0.04) | 65.48 (2.76) | 17.09 (0.64) | 68.80 (3.65) | 18.74 (0.68) |
| VAE-IAF | 88.32 (0.02) | 33.52 (0.02) | 87.91 (0.02) | 32.41 (0.02) | 75.31 (1.69) | 17.81 (0.73) | 76.11 (2.15) | 18.42 (0.66) |
| $\beta$-TC VAE | 90.96 (0.02) | 45.40 (0.05) | 91.91 (0.02) | **42.17 (0.07)** | 65.68 (0.91) | 24.14 (0.65) | 68.98 (2.67) | **25.57 (0.61)** |
| Factor VAE | 86.08 (0.06) | 31.38 (0.04) | 83.44 (0.05) | 31.76 (0.04) | 51.02 (1.73) | 15.77 (0.60) | 60.79 (2.06) | 17.56 (0.68) |
| InfoVAE-IMQ | 86.33 (0.04) | 32.48 (0.02) | 86.31 (0.06) | 32.10 (0.05) | 68.17 (2.34) | 16.65 (0.80) | 71.31 (2.62) | 18.10 (0.79) |
| InfoVAE-RBF | 85.94 (0.03) | 32.50 (0.03) | 86.12 (0.04) | 31.67 (0.03) | 66.02 (1.14) | 16.22 (0.69) | 71.93 (1.91) | 18.61 (0.67) |
| AAE | 93.28 (0.03) | 43.93 (0.07) | 94.31 (0.03) | 40.62 (0.11) | 74.19 (3.22) | 24.72 (0.75) | **80.41 (2.09)** | 24.76 (0.53) |
| MSSSIM-VAE | 78.30 (0.03) | 20.26 (0.06) | 76.54 (0.03) | 20.24 (0.04) | 49.33 (1.32) | 11.70 (0.19) | 48.58 (1.37) | 11.70 (0.17) |
| VAEGAN | 92.34 (0.02) | 26.56 (0.04) | 90.31 (0.03) | 29.90 (0.03) | **77.29 (1.19)** | 17.20 (0.45) | 79.67 (0.90) | 22.23 (0.44) |
| AE | 93.81 (0.02) | 42.15 (0.07) | 94.26 (0.03) | 40.47 (0.13) | 73.55 (0.60) | 23.19 (0.52) | 77.30 (0.84) | 23.18 (0.37) |
| WAE-IMQ | 93.60 (0.02) | **45.89 (0.07)** | 94.62 (0.03) | 41.35 (0.03) | 72.33 (2.92) | 23.81 (0.61) | 78.46 (3.48) | 25.09 (0.82) |
| WAE-RBF | 93.72 (0.02) | 43.38 (0.08) | 94.51 (0.02) | 40.63 (0.08) | 74.20 (1.94) | 23.70 (0.71) | 77.33 (1.92) | 24.66 (0.63) |
| VQVAE | 93.45 (0.02) | 42.89 (0.07) | **94.63 (0.04)** | 40.40 (0.09) | 72.61 (0.40) | 23.85 (0.48) | 76.68 (2.36) | 23.68 (0.37) |
| RAE-L2 | 94.75 (0.01) | 42.76 (0.08) | 94.43 (0.03) | 40.22 (0.05) | 74.07 (0.36) | 23.77 (0.54) | 78.66 (0.29) | 24.84 (0.73) |
| RAE-GP | **94.10 (0.02)** | 43.66 (0.07) | 94.45 (0.02) | 40.93 (0.14) | 72.88 (0.52) | **24.84 (0.53)** | 77.66 (1.29) | 23.86 (0.32) |

Table 3: Inception Score (higher is better) and FID (lower is better) computed with 10k samples on the test set. For each model and sampler we report the results obtained by the model achieving the lowest FID score on the validation set.

| Model | Sampler | ConvNet MNIST FID ↓ | IS ↑ | CIFAR10 FID | IS | CELEBA FID | IS ↑ | ResNet MNIST FID ↓ | IS ↑ | CIFAR10 FID | IS | CELEBA FID | IS |
|---|---|---|---|---|---|---|---|---|---|---|---|---|---|
| VAE | $\mathcal{N}$ | 28.5 | 2.1 | 241.0 | 2.2 | 54.8 | 1.9 | 31.3 | 2.0 | 181.7 | 2.5 | 66.6 | 1.6 |
|  | GMM | 26.9 | 2.1 | 235.9 | 2.3 | 52.4 | 1.9 | 32.3 | 2.1 | 179.7 | 2.5 | 63.0 | 1.7 |
| VAMP | VAMP | 64.2 | 2.0 | 329.0 | 1.5 | 56.0 | 1.9 | 34.5 | 2.1 | 181.9 | 2.5 | 67.2 | 1.6 |
| IWAE | $\mathcal{N}$ | 29.0 | 2.1 | 245.3 | 2.1 | 55.7 | 1.9 | 32.4 | 2.0 | 191.2 | 2.4 | 67.6 | 1.6 |
|  | GMM | 28.4 | 2.1 | 241.2 | 2.1 | 52.7 | 1.9 | 34.4 | 2.1 | 188.8 | 2.4 | 64.1 | 1.7 |
| VAE-lin NF | $\mathcal{N}$ | 29.3 | 2.1 | 240.3 | 2.1 | 56.5 | 1.9 | 32.5 | 2.0 | 185.5 | 2.4 | 67.1 | 1.6 |
|  | GMM | 28.4 | 2.1 | 237.0 | 2.2 | 53.3 | 1.9 | 33.1 | 2.1 | 183.1 | 2.5 | 62.8 | 1.7 |
| VAE-IAF | $\mathcal{N}$ | 27.5 | 2.1 | 236.0 | 2.2 | 55.4 | 1.9 | 30.6 | 2.0 | 183.6 | 2.5 | 66.2 | 1.6 |
|  | GMM | 27.0 | 2.1 | 235.4 | 2.2 | 53.6 | 1.9 | 32.2 | 2.1 | 180.8 | 2.5 | 62.7 | 1.7 |
| $\beta$-VAE | $\mathcal{N}$ | 21.4 | 2.1 | 115.4 | 3.6 | 56.1 | 1.9 | 19.1 | 2.0 | 124.9 | 3.4 | 65.9 | 1.6 |
|  | GMM | 9.2 | 2.2 | 92.2 | 3.9 | 51.7 | 1.9 | 11.4 | 2.1 | 112.6 | 3.6 | 59.3 | 1.7 |
| $\beta$-TC VAE | $\mathcal{N}$ | 21.3 | 2.1 | 116.6 | 2.8 | 55.7 | 1.8 | 20.7 | 2.0 | 125.8 | 3.4 | 65.9 | 1.6 |
|  | GMM | 11.6 | 2.2 | 89.3 | 4.1 | 51.8 | 1.9 | 13.3 | 2.1 | **106.5** | **3.7** | 59.3 | 1.7 |
| FactorVAE | $\mathcal{N}$ | 27.0 | 2.1 | 236.5 | 2.2 | 53.8 | 1.9 | 31.0 | 2.0 | 185.4 | 2.5 | 66.4 | 1.7 |
|  | GMM | 26.9 | 2.1 | 234.0 | 2.2 | 52.4 | 2.0 | 32.7 | 2.1 | 184.4 | 2.5 | 63.3 | 1.7 |
| InfoVAE - RBF | $\mathcal{N}$ | 27.5 | 2.1 | 235.2 | 2.1 | 55.5 | 1.9 | 31.1 | 2.0 | 182.8 | 2.5 | 66.5 | 1.6 |
|  | GMM | 26.7 | 2.1 | 230.4 | 2.2 | 52.7 | 1.9 | 32.3 | 2.1 | 179.5 | 2.5 | 62.8 | 1.7 |
| InfoVAE - IMQ | $\mathcal{N}$ | 28.3 | 2.1 | 233.8 | 2.2 | 56.7 | 1.9 | 31.0 | 2.0 | 182.4 | 2.5 | 66.4 | 1.6 |
|  | GMM | 27.7 | 2.1 | 231.9 | 2.2 | 53.7 | 1.9 | 32.8 | 2.1 | 180.7 | 2.6 | 62.3 | 1.7 |
| AAE | $\mathcal{N}$ | 16.8 | 2.2 | 139.9 | 2.6 | 59.9 | 1.8 | 19.1 | 2.1 | 164.9 | 2.4 | 64.8 | 1.7 |
|  | GMM | 9.3 | 2.2 | 92.1 | 3.8 | 53.9 | 2.0 | 11.1 | 2.1 | 118.5 | 3.5 | 58.7 | 1.8 |
| MSSSIM-VAE | $\mathcal{N}$ | 26.7 | 2.2 | 279.9 | 1.7 | 124.3 | 1.3 | 28.0 | 2.1 | 254.2 | 1.7 | 119.0 | 1.3 |
|  | GMM | 27.2 | 2.2 | 279.7 | 1.7 | 124.3 | 1.3 | 28.8 | 2.1 | 253.1 | 1.7 | 119.2 | 1.3 |
| VAEGAN | $\mathcal{N}$ | 8.7 | 2.2 | 199.5 | 2.2 | 39.7 | 1.9 | 12.8 | 2.2 | 198.7 | 2.2 | 122.8 | 2.0 |
|  | GMM | **6.3** | **2.2** | 197.5 | 2.1 | **35.6** | 1.8 | **6.5** | 2.2 | 188.2 | 2.6 | 84.3 | 1.7 |
| AE | $\mathcal{N}$ | 26.7 | 2.1 | 201.3 | 2.1 | 327.7 | 1.0 | 221.8 | 1.3 | 210.1 | 2.1 | 275.0 | 2.9 |
|  | GMM | 9.3 | 2.2 | 97.3 | 3.6 | 55.4 | 2.0 | 11.0 | 2.1 | 120.7 | 3.4 | **57.4** | 1.8 |
| WAE - RBF | $\mathcal{N}$ | 21.2 | 2.2 | 175.1 | 2.0 | 332.6 | 1.0 | 21.2 | 2.1 | 170.2 | 2.3 | 69.4 | 1.6 |
|  | GMM | 9.2 | 2.2 | 97.1 | 3.6 | 55.0 | 2.0 | 11.2 | 2.1 | 120.3 | 3.4 | 58.3 | 1.7 |
| WAE - IMQ | $\mathcal{N}$ | 18.9 | 2.2 | 164.4 | 2.2 | 64.6 | 1.7 | 20.3 | 2.1 | 150.7 | 2.5 | 67.1 | 1.6 |
|  | GMM | 8.6 | 2.2 | 96.5 | 3.6 | 51.7 | 2.0 | 11.2 | 2.1 | 119.0 | 3.5 | 57.7 | 1.8 |
| VQVAE | $\mathcal{N}$ | 28.2 | 2.0 | 152.2 | 2.0 | 306.9 | 1.0 | 170.7 | 1.6 | 195.7 | 1.9 | 140.3 | **2.2** |
|  | GMM | 9.1 | 2.2 | 95.2 | 3.7 | 51.6 | 2.0 | 10.7 | 2.1 | 120.1 | 3.4 | 57.9 | 1.8 |
| RAE-L2 | $\mathcal{N}$ | 25.0 | 2.0 | 156.1 | 2.6 | 86.1 | 2.8 | 63.3 | 2.2 | 170.9 | 2.2 | 168.7 | 3.1 |
|  | GMM | 9.1 | 2.2 | **85.3** | **3.9** | 55.2 | 1.9 | 11.5 | 2.1 | 122.5 | 3.4 | 58.3 | 1.8 |
| RAE - GP | $\mathcal{N}$ | 27.1 | 2.1 | 196.8 | 2.1 | 86.1 | **2.4** | 61.5 | 2.2 | 229.1 | 2.0 | 201.9 | 3.1 |
|  | GMM | 9.7 | 2.2 | 96.3 | 3.7 | 52.5 | 1.9 | 11.4 | 2.1 | 123.3 | 3.4 | 59.0 | 1.8 |

## Acknowledgments and Disclosure of Funding

The research leading to these results has received funding from the French government under management of Agence Nationale de la Recherche as part of the "Investissements d'avenir" program, reference ANR-19-P3IA-0001 (PRAIRIE 3IA Institute) and reference ANR-10-IAIHU-06 (Agence Nationale de la Recherche-10-IA Institut Hospitalo-Universitaire-6). This work was granted access to the HPC resources of IDRIS under the allocation AD011013517 made by GENCI (Grand Equipement National de Calcul Intensif).

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
