# OpenReview forum: "Pythae: Unifying Generative Autoencoders in Python - A Benchmarking Use Case"
_NeurIPS.cc/2022/Track/Datasets_and_Benchmarks — NeurIPS 2022 Datasets and Benchmarks _

### Official Review · Reviewer_31Cz · 2022-07-09
**Great effort, however requires validation of original work performance and comparisons on a higher resolution dataset**

**Rating:** 7
**Confidence:** 3

**Strengths:**

The main strength of this work comes from the proposed python library that should allow for a unified hub to compare different GAEs and find the best suited one for ones downstream task.

Authors tackle a crucial topic and compare performance of many GAEs under same settings. This is often neglected and hence it is not trivial to compare empirical performance of different methodologies in the literature.

Through comparative experiments, authors make valuable observations for the different tasks, especially for the image generation task.


**Weaknesses:**

In short, there are two major weaknesses; (i) it is not clarified whether the author implementations can replicate the performance of the original works, and (ii) benchmarks are conducted on simple, low resolution datasets, rendering conclusions of the manuscript unconvincing.

It has been seen in the computer vision community that findings in simple image domains (content and resolution) often do not translate well to more complex vision domains with higher resolution data. Authors conduct more than half of the experiments exclusively on MNIST and CIFAR10 datasets and derive conclusions based on their observations. Accordingly, these findings can be misleading and outright false for more complex datasets.
In Sec. 4.2.1 Task 2: Image generation, authors themselves also emphasize that using more advanced density estimation models as opposed to GMM was perhaps not useful because of the simplicity of the dataset they experimented with.

It would be vital to see if same results can be concluded after experimenting on ImageNet or other higher resolution natural datasets.

In Sec. 4.2.1, it is not clear how the latent dimensions are set for each dataset. For example, especially for CIFAR10, latent dimension of 256 for input images of 32x32 size seem unnecessarily high.

In Sec. 4.2.2, authors find out that often 16 or 512 latent dimensions were optimal depending on the task and/or GAEs. Unfortunately, these are boundary values within the search space the authors have used. Therefore, it is not right to conclude that 16 or 512 is optimal unless they observe the findings do not change after trying <16 and >512 dimensions.


**Additional Feedback:**

The authors accurately state that despite most recent works release their source codes, they are often not maintained. Given the fast changing scope of computing environments and updates in deep learning libraries that sometimes end up not being backward compatible, it would be encouraging to see a maintenance plan also being released with this manuscript.

**Clarity:**

Overall the paper is well written and reads well.
However, there are a few sentences that are not clear and require rephrasing. For example:
- lines 69-70: "pθ(x|z) is chosen depending [...]", not clear, please rephrase the whole sentence.

Authors should also fix typos. For example, see following lines;
- 57: decoder -> encoder
- 70: extra ")"
- 120: on-the-shelf -> off-the-shelf
- Table 2 caption does not define what is presented in parenthesis.

**Correctness:**

It is not clear whether the authors validated the experimental performance of their implementations of 21 GAEs with respect to the results presented in the official works. It is often a recurring problem that unofficial implementations, despite being very similar, do not manage to replicate the performance of the original work.
This reviewer believes that replicating the same performance as the original work is possibly more vital than the benchmarks conducted in the scope of this manuscript.


**Documentation:**

For reproducibility, I checked wandb.ai exports of the authors. It is possible to see the results of all conducted experiments there. Under https://wandb.ai/benchmark_team/trainings, it is possible to browse the configuration file used to run each experiment. Hence, it should be possible to reproduce the presented work.

**Ethics:**

There are no ethical concerns.

**Relation To Prior Work:**

The manuscript does not propose a new methodology, hence this is not applicable.

**Summary And Contributions:**

Authors propose Pythae, an open-source python library that implements 21 state-of-the-art generative autoencoders (GAEs). In addition, authors perform benchmarking of 19 GAEs on 5 downstream tasks using MNIST, CIFAR10 and CELEBA datasets. Throughout these tasks, authors deduce several conclusions on the benefits of different GAEs for different tasks over each other.

---

> ### Author Response · Authors · 2022-08-20
> **Thank you for your review!**
>
>
> Dear reviewer, thank you for your comments and mention of typos/mistakes which we have now fixed. Please find here a response to your concerns regarding weaknesses.
>
> **About original papers reproducibility**
> Thank you for this remark, this point was also noted by **R8G4x**. As a consequence, we have now validated our implementations by reproducing results from the papers when the official code has been released or when enough details about the experimental section of the papers were available (we indeed noted that in many papers key elements for reproducibility were missing such as the data split considered, which criteria is used to select the model on which the metrics are computed, the hyper-parameters are not fully disclosed or the network architectures is unclear making reproduction very hard if not impossible in certain cases), even if this sometimes required a substantial amount of work - code may be in a different framework, with unspecified or no longer supported dependencies making extra efforts needed to make the code work.
>
> This insists on the fact that the framework is flexible enough to reproduce results from publications. We now also provide the associated scripts on the github repo and trained models on the HuggingFace Hub (e.g. https://huggingface.co/clementchadebec/reproduced_iwae).
>
>
>
> **Benchmarks are conducted on simple, low resolution datasets.**
> As mentioned in the general response, the vocation of the benchmarking use-case isn’t to be as extensive as possible but rather to illustrate the capabilities of Pythae for benchmarking, with the example done on elementary datasets/tasks/metrics. This is why we only considered simple and widely used datasets.
>
> We agree that given more time and computation power higher resolution datasets would have been added to the benchmark. Indeed observed behaviors do not necessarily scale to more complex data. This is something we have tried to emphasize l.217 of the initial submission (l. 254 of the revision).
>
> **About the choice of the latent dimensions**
> The chosen latent dimensions each correspond to the ones where we observe the best global performances, although this choice is subject to interpretation as the “best” performance is hard to assess, we tried to pick the best compromise. We agree that 256 is surprisingly high for CIFAR10, and could have picked a lower dimension depending on what we judged to be the most important (see Figure 7 in appendix D).
>
> **About the *optimal* latent dimension**
> Agreed, these correspond to optimals within the range of explored latent dimensions which we did not explicitly specify, we have added this clarification in lines 301-302. The choice was made in this case to still limit ourselves to an exploration within the range of latent dimensions [16,512] as we do not wish to find an optimal latent space (which has no reason to be a power of 2) for each task/dataset but rather to illustrate that this optimal will change based on the chosen task and model type.
>
> ### Clarity
>
> Thank you for the feedback, changes have been made accordingly.
>
> ### Maintenance plan
>
> Thank you for this question ! The main author contact details will remain available and up-to-date on the gihub repo that will remain the main discussion channel. Moreover, we are currently thinking about adding back-up contributors that will also support this effort in the long-term.
>
> Do not hesitate if you have any other questions.
>
> Best,
>
> The authors

---

> > ### Comment · Reviewer_31Cz · 2022-08-24
> > **Thank you for your extra efforts, validation of most benchmarks brings great value to Pythae.**
> >
> > I understand your answer for my question regarding the simple datasets that were used for benchmarking purposes. Overall, I now agree that Pythae is within itself a good library for the community to try and experiment with different flavors of the AEs proposed in the recent years. Accordingly, I will increase my score.
> >
> > I am not fully convinced regarding the comparative findings and discussions done based on the simple datasets as well as the unreasonably high latent dimensions used in the trained models when reaching these conclusions. However I agree that it may not be a simple task and would require more computing resources. In summary, one should take Pythae as a library that implements the building blocks of different GAEs and not necessarily rely on the comparative discussions from the manuscript when deciding on the "best" flavor.
> >
> > Finally, as a form of pledge, please consider adding a section in the supplementary material discussing your maintenance plan.

---

> > > ### Author Response · Authors · 2022-08-25
> > > **Thank you for your positive feedback**
> > >
> > > Dear reviewer,
> > >
> > > Thank you very much for your positive feedback and considering increasing your score.
> > >
> > > We have in accordance added a maintenance plan to the supplementary material file.
> > >
> > > The choice of the "best" latent space for each dataset is indeed debatable, we had to settle on one per dataset in the main paper due to space constraints and readability reasons. Therefore we chose the latent dimension which seemed to us to be the best trade-off across models and tasks for latent spaces in [16, 32, 64, 128, 256, 512] according to Figure 7 of the *Appendix D.1 Effect of the latent dimension on the 4 tasks with the CIFAR10 database*. Moreover, that is true that there is often no clear consensus in the literature regarding the optimal latent dimension per dataset (for CIFAR10 in particular) and it is not rare to find quite high latent dimensions, especially in papers focusing on improving the generation from VAEs. For instance [1] uses latent code sizes of 128 (resp. 512) for CIFAR10 (resp. CELEBA), [2] uses latent dimensions of 512 for both dataset, [3] uses a latent code size of 128 for CIFAR10 and 64 for CELEBA while [4] and [5] consider a latent dimension of 512 and 128 for CIFAR10. Nonetheless, all of the presented experiments were also run on all latent space sizes within the set [16, 32, 64, 128, 256, 512]. Therefore, if needed anyone can have access to equivalents of Tables 1 and 2 in the paper for all latent dimensions and for the 4 tasks in wandb:
> > > - **reconstruction task**: https://wandb.ai/benchmark_team/reconstructions/
> > > - **generation task**: https://wandb.ai/benchmark_team/generations/
> > > - **classification task**: https://wandb.ai/benchmark_team/classifications/
> > > - **clustering task**: https://wandb.ai/benchmark_team/clustering/
> > >
> > > Please note that you can specify the model choice, database, latent dimensions and neural network architecture.
> > >
> > >
> > > Please also note that for the time being we do not see any change in your grade and hope you’ll be able to make the change by the end of the discussion round.
> > >
> > > Best,
> > >
> > > The authors
> > >
> > > [1] Parmar, G., Li, D., Lee, K., & Tu, Z. (2021). Dual contradistinctive generative autoencoder. In Proceedings of the IEEE/CVF Conference on Computer Vision and Pattern Recognition (pp. 823-832).
> > >
> > > [2] Pandey, K., Mukherjee, A., Rai, P., & Kumar, A. (2022). Diffusevae: Efficient, controllable and high-fidelity generation from low-dimensional latents. arXiv preprint arXiv:2201.00308.
> > >
> > > [3] Ghosh, P., Sajjadi, M. S., Vergari, A., Black, M., & Scholkopf, B. (2019, September). From Variational to Deterministic Autoencoders. In International Conference on Learning Representations.
> > >
> > > [4] Pandey, K., Mukherjee, A., Rai, P., & Kumar, A. (2021, September). VAEs meet Diffusion Models: Efficient and High-Fidelity Generation. In NeurIPS 2021 Workshop on Deep Generative Models and Downstream Applications.
> > >
> > > [5] Zhang, Z., Zhang, R., Li, Z., Bengio, Y., & Paull, L. (2020, November). Perceptual generative autoencoders. In International Conference on Machine Learning (pp. 11298-11306). PMLR.

---

> > > > ### Author Response · Authors · 2022-08-29
> > > > **Kind reminder: The discussion period is about to end**
> > > >
> > > > Dear reviewer,
> > > >
> > > > We feel a little uncomfortable asking you this again, but we still don't see any change in your score (still at 5) when you said in your previous comment that you were going to increase it.  Since the discussion period is going to end in a few hours and this is an important change for the paper, we just wanted to give you a little reminder about this.
> > > >
> > > > We thank you again for taking the time to review the document and respond to the rebuttal.
> > > >
> > > > Best,
> > > >
> > > > The authors

---

### Official Review · Reviewer_8G4x · 2022-07-26
**A Python environment for autoencoders that should have a clearer discussion of prior works and have a more nuanced experimental assessment**

**Rating:** 7
**Confidence:** 5

**Strengths:**

There are various strengths to the proposes Pythae environment , primarily perhaps that it comes with a promise of providing a framework to easily experiment and compare various autoencoder based approaches.

* Having the Pythae tool holds a promise towards more reproducible research (although simultaneously see first weakness below)
* Although the considered approaches are all autoencoders, the breath of considered methods is impressive. There are of course a couple of examples that may be still added, but as the authors say the library should be subject to continuous development and as such will likely only grow to be more exhaustive over time.
* The supplementary material overall is very helpful. Similarly, the usage examples and readme instructions in the provided GitHub repo are pretty comprehensive and are likely to facilitate adoption.
* Appendix section D.4 is much appreciated, in particular, the effort to provide a short summary of the main mathematical advance in each respectively model.

**Weaknesses:**

* As mentioned in the first point on strengths, the work holds a promise towards reproducing existing works. At the same time, when looking at the provided experimentation in the paper, some of the choices are a bit puzzling in this regard. For instance, there does not seem to be any mention or measuring of e.g. Kullback-Leibler (KL) divergences in any of the variational approaches. Similarly, reconstruction loss seems to be measured in a mean squared error, rather than what is typically reported in almost any generative modeling paper. It’s not clear to me why the choice has been made here to deviate from the by now fairly standardized convention to stick to reporting of (negative) log likelihoods. Naturally, this does not hinder the relative comparison between methods, but it does make it harder to assess the correctness with respect to the original papers and hinders their direct comparison/reproduction a bit. This points feels particularly important as the scale of values (MSE lying around 0.01 and log likelihoods typically being multiple orders of magnitude larger) can significantly affect choice of hyper-parameters etc.
* The discussion of “enhancing the model” of section 2.2 is rather shallow. I understand the space constraint here, but it feels like crucial and heavily investigated arguments are missing. For example, any mention with respect to desired/undesired prior-posterior mismatches or any discussion on lossy compression is omitted here in favor of a too simplified narrative.  This is a bit problematic because the “disentanglement” picture is rather naive and it is unclear whether it is an agreed upon perspective.  There have been several well cited published papers, to name a few: “Disentangling disentanglement in variational autoencoders”, “resampled priors for variational autoencoders”, “rethinking lossy compression: the rate-distortion-perception tradeoff”, that challenge the narrative that autoencoders and variational versions of it are all about finding a better form for the posterior or that a weighting of the KL term (beta-vae) somehow induces disentanglement (in whichever way disentanglement is actually defined).
* Following up on the above two arguments, I am a bit worried that the experimental evaluation may be somewhat misleading to the reader. In particular, the fact that the tasks are separated in the way they are does not seem very intuitive to me. For instance, what is the purpose of running an experiment measuring MSE reconstruction I na comparison between an AE, a VAE, and a beta-VAE and then concluding that the beta-VAE is a better model if one simply turns down the divergence term? Similarly, task 3 does seem to be somewhat oddly formulated and catered heavily towards the AE again.  If we are interested in classification, why first do unsupervised pre-training and then train a linear classifier on top. Naturally the space of auto-encoders is much less constrained than that of any VAE, but what we would actually be interested in seems to be learning the joint p(x,y), rather than freezing the architecture. (Like in the semi-supervised variants of Kingma et al). As a third point, there should be a discussion around FID and it’s brittleness with respect to how useful it is as a measure (see e.g. paper “pros and cons of GAN evaluation measures” as one example). Why would we be interested in measuring an inception distance, which is imagenet based,  if we train a model on gray-scale or binarized handwritten digits? Finally, perhaps the individual investigations are not troublesome in the sense that they were conducted, but the way that they are described, in a seemingly constant attempt to select “a clear winner per category” is rather misleading to the reader. Evaluation of generative models has been an ongoing debate for several years now and it is clear that is complex and challenging. It would be great to have some of this flavor depicted in the experimentation and discussion, rather than going for an overly simplified take-away.
* See related work section below, on the library not being the first of its kind and lacking a comparison

**Additional Feedback:**

As the paper and repo are open source and non-anonymous (and have in fact been so for a while it seems), I wonder why line 154-155 says that trained models are intended to be shared solely upon acceptance.

**Clarity:**

The main part about clarity is summarized in the last point in the weakness section. Apart from this, I am curious why table 2 has included a measure of statistical deviation across random seeds, but tables 1 and 3 seem to be depicting single values.

**Correctness:**

* Probably just an accidental mistake, but lines 53 to 57 first call p(z|x) the decoder and then refer to the variational distribution q(z|x) as the decoder again, which should be the encoder instead.
* Lines 84-86 say “the ELBO objective can indeed be rewritten as the difference between the true log probability and a KL divergence between the approximate posterior and the true posterior”. Perhaps tis is just a wording issue, but the argument somehow sounds “reversed” to me. The ELBO initially arises from the fact that we cannot compute the KL divergence to the true posterior, which is why we construct the lower bound, since a KLD is strictly positive.
* The word “proven” is being used in a sense that is misleading or untrue in various places. In a paper speaking of generative models and variational inference, the word proven should be reserved for ideas that are actually theoretically proven and not just empirically demonstrated. Specifically: lines 3, 28, 176

**Documentation:**

The documentation is sufficient. In particular, the usage example, thorough readme in the GitHub repo and detailed supplementary descriptions are useful.

**Ethics:**

There are no ethical considerations

**Relation To Prior Work:**

Prior works has been discussed in adequate detail with respect to each individual model (especially in the summaries in the supplementary material). On the overall motivation side, it can/should be improved, as denoted in above weaknesses section.

Apart from the technical and mathematical side of descriptions, I am however wondering if Pythae indeed is the first attempt at consolidating autoencoder implementations, as suggested by the paper. Surely there exist several prior coding efforts that could be attributed with the credit they deserve. I am leaving this as a partial question, but also am aware that at least e.g. this repository exists: [GitHub - AntixK/PyTorch-VAE: A Collection of Variational Autoencoders (VAE) in PyTorch.](https://github.com/AntixK/PyTorch-VAE) With roughly 700 forks, 3.6k stars and 18 implemented autoencoder variants (some of which are the same, some of which are different from Pythae), I think the paper should really provide a more in-depth discussion of the differences, and in particular some credit attribution to prior attempts. Is Pythae mainly about improving the implementations? Is it about the evaluation being streamlined? Some differentiation would be appreciated.


**Summary And Contributions:**

The paper proposes Pythae, a Python library that provides implementations for various types of popular autoencoder (AE) architectures and modeling choices. Beside an introductory description of the library and a summary of each individually considered AE (also in supplementary pdf), 19 different AEs are benchmarked on various tasks ranging form measuring reconstruction losses to proxies to assess generation quality.

POST DISCUSSION UPDATE:
I believe many of my concerns have been addressed, improvements have been made and more outlined for a camera ready version. As detailed in the response below I encourage the authors to continue tuning the presentation and am raising my rating to recommend acceptance of the paper.  (Previously 5 to now 7)

---

> ### Author Response · Authors · 2022-08-20
> **Thank you for your review! [1/3]**
>
> Dear Reviewer, thank you for your careful review of our paper and your detailed feedback and suggestions to improve it. In particular, thank you for noticing the effort behind Appendix section D.4. Please see below some clarifications on the weaknesses you pointed out.
>
> - **Puzzling choices for experimentation**: As explained in the general comment, the main purpose of the benchmarking use case is to support the claims about the ease of use and flexibility of the framework, and is meant to explicit trends rather than judge performance. This is why we consider elementary tasks, datasets and metrics. We agree that the KL divergences or the NLL are commonly used metrics for variational approaches, however they do not adapt to non variational models, which is why we only use metrics that are suitable for all the models we consider such as the MSE or FID.
>
>     We nonetheless understand that since MSE is rarely mentioned as a metric for variational based methods, it makes it hard to assess the correctness of the implementations when compared to original publications.
>
>     As for reproducing work from original publications, this point was also noted by **Reviewer 31Cz**. As a consequence, we have now validated our implementations by reproducing key results from each paper when the official code has been released or when enough details about the experimental section of the papers were available (we indeed noted that in many papers key elements for reproducibility were missing such as the data split considered, which criteria is used to select the model on which the metrics are computed, the hyper-parameters are not fully disclosed or the network architectures is unclear making reproduction very hard if not impossible in certain cases), even if this sometimes required a substantial amount of work - code may be in a different framework, with unspecified or no longer supported dependencies making extra efforts needed to make the code work.
>
> - **About prior works**: The original purpose of section 2.2 was both to introduce and justify our choice of the models implemented in Pythae. We understand that the title of this section may have been unclear and we have now changed it to “Improvements upon the classical VAE method”, and agree that challenging the presented axes of improvement of VAEs would notably enrich this section. We have taken this into consideration and added a part challenging the prior/posterior mismatch issue (l.78 to 82), as well as a narrative challenging the different disentanglement approaches and adding more nuance to its interpretation in lines 102 to 115. However, we wish not to delve too deep into the pros and cons of each approach as we consider this to be outside of the scope of this section.
>
>     Additionally, could you be more specific on what you mean by a discussion on lossy compression as we do not see how it would fit inside the scope of this section ?

---

> ### Author Response · Authors · 2022-08-20
> **Thank you for your review! [2/3]**
>
> - **On the experiemental section**: As explained in the general comment, we wish to reveal trends and behaviours rather than judge performances, as it is hard to find a metric or task that would be fair to all models. The tasks were therefore chosen as we consider them to be elementary and correlated to desired behaviours of generative autoencoders (capacity of reconstruction, generation, “quality” of the latent space representation ...). This point has now been clarified in section 4.
>
>     We agree that task 1 is favoured towards AEs, and the same could be argued about task 3 (although the classical VAE prior might be unfavorable towards latent classification, this is not necessarily true for all variational based models, for example, VAMP learns a multimodal prior based on the training dataset and achieves performances comparable to AE methods), which is what we wish to explore). Indeed all tasks are chosen to cover one aspect of GAEs, which is why by nature of construction tasks 1 and 3 tend to favour AEs, while tasks 2, 4 and 5 will favour variational methods - the point here is not to pick a winner but rather to see if general trends emerge, and also to test the capabilities/versatility of models.
>
>     Task 1 therefore seemed like a must have, even if the results were foreseeable. For task 3 we do not want to find the best possible performances on classification, which would indeed be achieved by learning the joint p(x,y). As explained in line 226, to highlight the meaningfulness of the learned latent representations we perform a simple classification task with a single layer classifier as explained and proposed in the method of [1]. This method was for instance also used in [2] to quantify the quality of the latent representations learned by their model.
>
>     We agree that a discussion should be added on the interpretability of FID and IS, especially in the case of the MNIST dataset, in consequence we have mentionned and referenced debates regarding FID/IS in lines 246-248. Nonetheless, it should be noted that using the FID metric on MNIST is also commonly used e.g. [3], [4]
>
>
> [1] Adam Coates, Andrew Ng, and Honglak Lee. An analysis of single-layer networks in unsupervised feature learning. In Proceedings of the Fourteenth International Conference on Artificial Intelligence and Statistics, 2011.
>
> [2] David Berthelot, Colin Raffel, Aurko Roy, and Ian Goodfellow. Understanding and improving interpolation in autoencoders via an adversarial regularizer. In International Conference on Learning Representations, 2019
>
> [3] Partha Ghosh, Mehdi SM Sajjadi, Antonio Vergari, Michael Black, and Bernhard Schölkopf. From variational to deterministic autoencoders. In 8th International Conference on Learning Representations, ICLR 2020, 2020.
>
> [4] Bin Dai and David Wipf. Diagnosing and Enhancing VAE Models. In International Conference on Learning Representations, 2018

---

> ### Author Response · Authors · 2022-08-20
> **Thank you for your review! [3/3]**
>
> - **As to libraries proposing VAE implementations**: We agree with the reviewer that there are other noteworthy open-source libraries providing VAE-based model implementations: GitHub - AntixK/PyTorch-VAE: A Collection of Variational Autoencoders (VAE) in PyTorch. is indeed a good example.
> Pytorch-VAE is a great repository that gathers VAE implementations proposed in publications, but it only provides the user with scripts allowing to train the model as presented in the original paper. This code was indeed not initially designed to allow for dataset or network changes but rather with a focus on sticking to the model proposed in the original publication. This was for instance noted and requested by a user in this [issue](https://github.com/AntixK/PyTorch-VAE/issues/52#issue-1153430613). As a result, providing flexibility was both the main motivation behind Pythae and the main constraint driving its design. However, citing properly this repository was indeed missing and we have now added the reference accordingly in the introduction of section 3.
>
>     Moreover, we believe Pythae is both useful and needed, as it distinguishes itself from other existing libraries for the following reasons:
>
>     - The modular structure of Pythae provides the possibility to combine sampling methods with different models, easily change model hyper-parameters, training configurations and neural networks.
>     - We propose a library and not only a repository that can be installed with pip and conda and allows easy integration in other codes. This also allows to have a code versioning and clear releases to make the code robust and stable at any time.
>     - We provide documentation (which was deemed “very clear” by **Reviewer aVDB**) and tutorials to lower the entry barrier.We also included an experiment tracking tool that we hope will reveal itself to be particularly useful to compare models or training runs.
>     - We now have also validated our implementations with reproduction of experiments proposed in the original paper when possible.
>     - The library now integrates (this was done during the reviewing process) an online model sharing tool, the HuggingFace Hub, allowing to share models with peers
>     - Pythae contains implementations of models that are not implemented in Pytorch-VAE or other libraries.
>
>
> We believe that these libraries target different goals and can complete themselves.
>
>
> ### Correctness
>
> - **Q**: Probably just an accidental mistake, but lines 53 to 57 first call p(z|x) the decoder and then refer to the variational distribution q(z|x) as the decoder again, which should be the encoder instead.
>
>     **A**: Thank you, this has been corrected
>
> - **Q**: Lines 84-86 say “the ELBO objective can indeed be rewritten as the difference between the true log probability and a KL divergence between the approximate posterior and the true posterior”. Perhaps tis is just a wording issue, but the argument somehow sounds “reversed” to me. The ELBO initially arises from the fact that we cannot compute the KL divergence to the true posterior, which is why we construct the lower bound, since a KLD is strictly positive.
>
>     **A**: Agreed, we used rewritten in the sense that we first introduce the ELBO objective in our paper using the expectation over the latent variables, then re-introduce the ELBO with another way of writing it. We now replaced the word rewritten with written for coherence’s sake.
>
> - **Q**: The word “proven[SA14] ” is being used in a sense that is misleading or untrue in various places. In a paper speaking of generative models and variational inference, the word proven should be reserved for ideas that are actually theoretically proven and not just empirically demonstrated. Specifically: lines 3, 28, 176
>
>     **A**: The word proven in these cases is used within the expression “proven to be” which we use in the sense “has shown itself to be” (empirically). Changed in lines 28, 202 (previously l176) for disambiguation.
>
>
> ### Additional Feedback
>
> - **Q**: As the paper and repo are open source and non-anonymous (and have in fact been so for a while it seems), I wonder why line 154-155 says that trained models are intended to be shared solely upon acceptance.
>
>     **A**: Due to the number of trained models (thousands), we have decided not to release the trained models since we considered their relevance rather limited. We rather decided to share the models obtained when reproducing the results in the initial publications.
>
>
> Do not hesitate if you have any other questions,
>
> Best,
>
> The authors

---

> > ### Comment · Reviewer_8G4x · 2022-08-29
> > **Additions have largely addressed feedback and have promised some further improvements**
> >
> > I thank the authors for their detailed responses, for taking into account my feedback, and for starting to include some of the corresponding revisions.
> >
> > In particular, I much appreciate the reproduction of NLLs and related metrics for various models. I believe the paper and tool is much stronger with such shown reproduction capacity, as many available implementations online fail to achieve the latter and it is often times hard to find out why.
> > Similarly, the addition of a maintenance plan, as suggested by one of the other reviewers, and the revised softer wordings are appreciated and improve the paper. Many of my concerns have been addressed and I hope the remaining ones will be similarly included in a final camera ready version with more time.
> >
> > As a consequence, I am raising my rating to recommend acceptance. However, I would like to note that there are some suggested changes and aspects in the response that I presently was not yet able to find in the actual revised pdf. I still believe that section four would benefit from a much stronger wording/discussion to clarify the intent of the benchmark to the reader more carefully (to avoid the mentioned pitfall of thinking that it is about "selection the best model", as denoted in the response of the authors) and the addition of the reproducibility parts to the main paper would be valuable as well.
> >
> > I assume and strongly encourage the authors to make these additional changes, finish up the extra experimental values and provide the respective revisions in form of  text restructuring for the discussion in the final camera ready.

---

> > > ### Author Response · Authors · 2022-08-29
> > > **Thank you for your positive feedback**
> > >
> > > Dear reviewer,
> > >
> > > Thank you for taking the time to read our rebuttal, your positive feedback and for increasing your score.
> > >
> > > We will work to add the remaining suggested changes and we will add them to the camera-ready version if this paper is accepted. In particular, we will polish section 4 to better position the benchmark objective and include the reproducibility experiments in the main paper (or in the appendices due to space constraints in the main article).
> > >
> > > Best,
> > >
> > > The authors

---

### Official Review · Reviewer_aVDB · 2022-07-26
**A well-documented toolbox for training, analyzing and evaluating various geneative autoencoder models.**

**Rating:** 7
**Confidence:** 3
**Correctness:** Yes.

**Strengths:**

1. This paper is overall well-written and the documentation in the open-source library is very clear.
2. Besides a clear introduction to the code structure and project, the authors also extensively compare various AE variants from different aspects, including image reconstruction and generation, latent vector classification and clustering, and image interpolation.
3. Experiment results also provide insight into how different components used in different AE models affect the five aspects studied.


**Weaknesses:**

1. Experiments are performed on relatively small-scale datasets like MNIST and CIFAR. Experimenting on larger-scale datasets like ImageNet would better illustrate the efficacy of the proposed toolbox.
2. Many recent unimodal/multimodal pretrained models  (e.g., BEIT, DaLLE)for text-conditioned image/video generation are also based on VQVAE. It would be interesting to also include some results on VAE used in these pretrained models.

 [1] BEiT: BERT Pre-Training of Image Transformers, ICLR, 2022
 [2] DALL·E: Creating Images from Text, 2021.

**Additional Feedback:**

Please see the "weakness" section.

**Clarity:**

This paper is well written and easy to follow.


**Documentation:**

The documentation of the toolbox is clear.

**Ethics:**

I don't see any problems.


**Relation To Prior Work:**

There are some discussion on prior AE models.

**Summary And Contributions:**

This paper provides a python toolbox called Pythae to train and evaluate various AE models under a unified framework. This library has aroused vast interest among related users. In particular, there have already been over 800 stars in the open-source repository. This work
also extensively compare various AE models from various aspects and provide insightful dicussions.

---

> ### Author Response · Authors · 2022-08-20
> **Thank you for your review!**
>
> Dear Reviewer, thank you for taking the time to review our paper and for your positive feedback.
>
> We are grateful that you recognize the amount of work behind this framework and noted that it has already raised some marked interest in the community. We indeed put a particular care on the design of the library and chose to add clear documentation and tutorials to make it as accessible and flexible as possible.
> We are also very happy to see that external users have already appropriated the library, requested new models/features (see [issues](https://github.com/clementchadebec/benchmark_VAE/issues)) and that some have also contributed directly to it. Please also note that during the reviewing period we have also integrated a model sharing tool (HuggingFace Hub) to share trained models with peers.
>
> - **W1**: We agree that given more time and computation power IMAGENET would have been added to the benchmark and support that Pythae also handles this type of data. Nonetheless, we designed the toolbox so it is as flexible as possible regardless of the data type that is provided to the model. For instance, we already have had some users using the library on other modalities different from images (see for instance https://github.com/clementchadebec/benchmark_VAE/issues/36).
>
> - **W2**: We thank the reviewer for this suggestion, this library is intended to evolve in time and the integration of unimodal/multi-modal is something we may consider adding to Pythae in the future.
>
> Do not hesitate if you have any other questions.
>
> Best,
>
> The authors

---

### Official Review · Reviewer_NiL8 · 2022-07-29
**Review of Pythae**

**Rating:** 6
**Confidence:** 4
**Clarity:** The paper is well-written and well-or…

**Strengths:**

- **{S1}** The framework enables researches to train (custom) VAE-based models in few lines of code, and thus reduces friction for further research in this area.
- **{S2}** The framework seems to be well-documented and is readily available.
- **{S3}** The experimental section in the paper is interesting, especially the part about varying sizes of latent spaces. However, see {W1}.

**Weaknesses:**

- **{W1}** My main concern is that the experimental section is evaluated with IS/FID that use Inception as feature extractor. Several works address problems with these metrics, e.g. see [1-5]. If time permits I would suggest to run more metrics (e.g. [4] or [5]) and additionally report those in the paper. Otherwise, at least mentioning these concerns, suggesting additional alternatives, and stating that benchmarking on IS/FID alone is insufficient should be mandatory.
- **{W2}** The linked framework does not seem to contain the code of the experimental section in the paper, i.e. benchmarking different VAEs and implementations of metrics like FID/IS. Please clarify if this is the case or if I was not able to find it.
- **{W3}** In relation to {W2}: The submission is neither a dataset, nor a benchmark(ing tool), but a framework to train (custom) VAE-based generative models in few lines of code. I'm unsure whether this fits the scope of the dataset and benchmark track.

* [1] A Note on the Inception Score, https://arxiv.org/abs/1801.01973
* [2] Effectively Unbiased FID and Inception Score and where to find them, https://arxiv.org/abs/1911.07023
* [3] Internalized Biases in Fréchet Inception Distance, https://openreview.net/forum?id=mLG96UpmbYz
* [4] On Self-Supervised Image Representations for GAN Evaluation, https://openreview.net/forum?id=NeRdBeTionN
* [5] The Role of ImageNet Classes in Fréchet Inception Distance, https://arxiv.org/abs/2203.06026

**Additional Feedback:**

The references to the Appendix contain a fullstop (e.g. L187). Is this intentional?

**Correctness:**

The submission is neither a dataset, nor a benchmark, but a framework to train (custom) VAE-based generative models in few lines of code. See {W3}.

**Documentation:**

The framework is available on GitHub, contains tutorials/examples, and a documentation on readthedocs.io.

**Ethics:**

I do not see any ethical concerns with this paper.

**Relation To Prior Work:**

The motivation for this framework is that no common framework exists that combines code for different concepts of VAE-based models, hindering research in this area.

**Summary And Contributions:**

This paper introduces an extendable framework to train different VAE-based generative models. A use-case is presented benchmarking 19 different autoencoders on 5 different tasks (image reconstruction, image generation, classification, clustering, interpolation).

---

> ### Author Response · Authors · 2022-08-20
> **Thank you for your review!**
>
> Dear reviewer, we would like to first thank you for taking the time to review our paper. Your comments have been taken into consideration and have allowed us to update and improve our paper. Please find below our clarifications on the issues you have pointed out.
>
> - **W1** : As explained in the general comment, the core contribution of this paper is the open-source library Pythae. The main goal of the benchmark is therefore to illustrate and support the claims about the flexibility and ease of use of the code, and results are meant to explicit common behaviours rather than solely compare model performances. This is why we only consider elementary tasks with commonly used metrics and datasets for this benchmark. We agree that if the main contribution of the paper had been a performance benchmark, a more thorough investigation on the choice of metrics would have been necessary. Although using FID on a dataset has its limitations (especially in the case of datasets with non-natural images), it is still widely used in the community which is why we decided to use this metric in the benchmarking use case. As you suggested, we now better emphasize that the metrics used in the paper have some limitations and have adapted the analysis of the results in consequence in section 4.
>
> - **W2**: Thank you for highlighting this point, we planned on giving access to the scripts upon request but have now decided to add the scripts to the supplementary materials. As you can see these scripts again show the flexibility provided by the framework that allows model training and generation in a few lines of code. As to the metric computation, for the FID and IS we used the official implementations with the associated github repositories referenced in the footnotes of the paper.
>
> - **W3**: The section describing the scope of submissions of the NeurIPS 2022 Datasets and Benchmarks Track states “Data generators, reinforcement learning environments, or benchmarking tools are also in scope.” which is why we consider our paper to be within the scope of submission.” We agree that this submission is neither a dataset nor only a benchmark, it is an open-source library which can be used:
>
>     - As a data generator: Through the pythae.samplers module, data can seamlessly be generated and reused for downstream application such as data augmentation
>
>     - As a benchmarking tool: We aim to show that thanks to its modular structure, the library could reveal itself to be very useful for (but not limited to) benchmarking applications.
>
>     - As a reproducible research environment: We also think that beyond the benchmarking aspect, gathering models under a unified implementation, training them with a unified framework, unified training conditions and network architectures allows straightforward reproducibility (please also note that following this discussion we have reproduced the results presented in papers when possible : results from papers having either an open-source implementation or providing enough experimental details about the model they used in their experiments were all reproduced). This insists on the fact that the framework is flexible enough to reproduce results from publications. We now also provide the associated scripts on the github repo and trained models on the HuggingFace Hub.
>
> Do not hesitate if you have any other questions.
>
> Best,
>
> The authors

---

> > ### Comment · Reviewer_NiL8 · 2022-08-29
> > **Score Update**
> >
> > Dear Authors,
> >
> > Thank you for considering my suggestions, especially addressing {W1} and {W2}. I updated my score accordingly. As to {W3}, I still remain unsure whether Pythae fits the scope of this track. Following the given justification, any library training and evaluating models (e.g. PyTorch itself) could be considered in scope.

---

> > > ### Author Response · Authors · 2022-08-29
> > > **Thank you for your positive feedback.**
> > >
> > > Dear reviewer,
> > >
> > > Thank you for taking the time to read our rebuttal, for your positive feedback, and for increasing your score. We are glad that you saw the utility of Pythae as an easy-to-use library to enable the acceleration of VAE based researches.
> > >
> > > Considering the scope of this track, from what we understand, the purpose of this track is to promote, among other things, benchmarking tools useful for future research experiments which is what we try to highlight in this paper. Moreover, these libraries are difficult to be submitted to other conferences because they cannot be easily anonymized. We also wanted to remember that our primary motivation for developing and designing Pythae was to have an easy-to-use tool for comparing apples to apples. That is why we think that the library can be seen as a benchmarking tool, even if it can be used for other applications as well.
> > >
> > > Best,
> > >
> > > The authors

---

### Author Response · Authors · 2022-08-20
**Thank you for your reviews! Some clarifications [1/2]**

First of all, we would like to sincerely thank the reviewers for taking the time to carefully read and review our paper as well as the open-source library, and for the useful feedback which has allowed us to improve and revise our article in accordance with the given comments.

We will address the individual concerns raised by the reviewers in separate comments and provide here a global response to better clarify some points:

### Focus of the paper
We would like to insist on the fact that the core contribution of this paper is the presentation of the Pythae library. Although the benchmarking case study leads to some interesting findings, it is meant as an illustration of the capabilities of the library. We have clarified this point in the paper by insisting more on the presentation of the library (line 43 in Section 1, Section 3 has been rewritten with a clear emphasis on the vision and advantages of Pythae, introduction of Section 4.), as well as insisting on the illustrative aspect of the benchmark and alleviating its conclusions (Introduction of Section 4, conclusions on the benchmark have been removed from section 5.)

### The Pythae library
The presentation of Pythae has been modified to better explain the following points. Pythae is designed with the following points in mind:
-  **Usable by all**: Pythae makes GAE models accessible to all - beginners to experts. This means beginners can run *ready-to-use* models with a few lines of code, while more advanced users can easily access and adapt different methods to their specific use-cases, with custom encoder/decoder definition. Indeed, the library was designed to be flexible enough to allow users to use existing implementations on their own data, with custom model hyper-parameters, training configurations and network architectures. The library has an online documentation (full documentation can be found at https://pythae.readthedocs.io/en/latest/) and is also explained and illustrated through tutorials available either on a local machine or on the *Google Colab* platform.

- **Unified implementation**: The brick-like structure of Pythae allows for seamless but efficient interchange between models, sampling techniques, network architectures, model hyper-parameters and training schemes. Pythae is unit-tested ensuring code quality and continuous development with a code coverage of 98% as of release 0.6. The library is made available on *pip* and *conda* allowing an easy integration. Its development is performed through releases that ensure stable and robust implementations.

- **A reproducible research environment**: Pythae is open to all and as such encourages transparent and reproducible research, as illustrated in the next section. With a variety of different interchangeable models gathered in a common library, it can be used as a sandbox for research and applications. Moreover, the library also integrates an easy-to-use experiment tracking tool (*wandb*) allowing to monitor runs launched with Pythae and compare them through a graphic interface, and an online model sharing tool, the HuggingFace Hub, allowing to share models with peers.

- **Evolving and driven by the community**: Pythae's design is intended to evolve with the addition of new models to enrich the existing model base. Furthermore, peers can contribute by reviewing and submitting models to enrich the library, a few of which have already been added at the time of this publication.

We are glad to see it has already raised some interest in the community with >840 stars, ~ 2.9k downloads (2 months after the official release), while new users have started appropriating the library, requesting new features (see [issues](https://github.com/clementchadebec/benchmark_VAE/issues)) and some have already contributed. This first illustrates the rising interest and need for this unified framework but also correlates with our vision to see Pythae grow thanks to peer contributions and will allow more reliable implementations since we expect the community to catch potential bugs.

---

### Author Response · Authors · 2022-08-20
**Some clarifications [2/2]**


### Reproducing results from other papers
As pointed by **Reviewer 31Cz**, the reproducibility of the original papers results is an important validation of Pythae implementation and we thank the Rewiewer for pointing this out. Therefore, we have been able to reproduce the results of most of the 21 models existing in the library, with the exception of papers where neither open access source code nor comprehensive indications on implementation details are available.
We have open-sourced the scripts, configurations and results on the repository at https://github.com/clementchadebec/benchmark_VAE/tree/main/examples/scripts/reproducibility and made the trained models available on the HuggingFace Hub (e.g. https://huggingface.co/clementchadebec/reproduced_iwae).


### The benchmark
The benchmarking itself does not intend to find the best model, as this comparison would unfairly favour certain models, but rather to show that comparison between models can easily be performed using the library, on any given task, whether this involves using a simple metric (mse) or a more elaborate one (clustering on the latent space). The choice was therefore made to compare models on elementary tasks, datasets and metrics. The introduction and conclusions in section 4 have been modified in order to better underline this point, and the script allowing to generate the benchmark has been added in the supplementary materials.

Following this point, the benchmark’s aim is to underline common behaviours rather than judge performances, which can be subject to hyperparameter biases. The key findings therefore are meant to be focused solely on general trends within groupings of models, which are less dependent on possible biases, and are purposely given without strong interpretations or conclusions. We agree that sentences that focus more on individual performances should therefore be deleted from our conclusions, and both introductions and conclusions in section 4 have been modified in accordance with your reviews.


| Model | Dataset | Metric | Obtained value | Reference value | Note|
|---|---|---|---|---|---|
| VAE | Binary MNIST | NLL (200 IS) | 89.78 (0.01) | 89.9 | using [1] |
| VAMP (K=500) | Binary MNIST | NLL (5000 IS) | 85.79 (0.00) | 85.57 |
| SVAE | Dyn. Binarized MNIST | NLL (500 IS) | 93.27 (0.69) | 93.16 (0.31) |
| IWAE (n_samples=50) | Binary MNIST | NLL (5000 IS) | 86.82 (0.01) | 87.1 |
| HVAE (n_lf=4) | Binary MNIST | NLL (1000 IS) | 86.21 (0.01) | 86.40 |
| BetaTCVAE | DSPRITES | Modified ELBO/ELBO (after 50 epochs) | 710.41/85.54 | 712.26/86.40 |
| RAE_L2 | MNIST | FID | 9.1 | 9.9 |
| RAE_GP | MNIST | FID | 9.7 | 9.4 |
| WAE | CELEBA 64 | FID | 56.5 | 55 |
| AAE | CELEBA 64 | FID | 43.3 | 42 |

*Note*:
We deemed BetaVAE reproduced since our VAE implementation seems valid.
For VAEGAN no generative metric is provided but generated images are quite similar to the one shown in the paper (easily recognizable with the adversarial approach). We are the authors of the RHVAE paper which is considered in line with the official code.


Do not hesitate if you any other questions,

Best,

The authors

[1] Danilo Rezende and Shakir Mohamed. Variational inference with normalizing flows. In International Conference on Machine Learning, pages 1530–1538. PMLR, 2015


-------------------------------------------------------- Edit 25/08/22 --------------------------------------------------------

Further to **reviewer 31Cz** feedback, we have added a maintenance plan to Appendix. A of the supplementary material pdf.

---

### Author Response · Authors · 2022-08-29
**Kind reminder: The discussion period is about to end**

Dear Reviewers NiL8, aVDB, 8G4x,

We are taking the liberty of contacting you again because the discussion phase between authors and reviewers will end in 12 hours (1 PDT on Monday 29) and we wanted to make sure that you are happy with the answers to your reviews and do not have any additional questions regarding our paper or rebuttal.

We thank you again for taking the time to read and review our work and for your useful feedback.

Best,

The authors

---

### Meta-Review · Area_Chair_jTNk · 2022-09-10

**Recommendation:** Accept
**Confidence:** 4

**Metareview:**

Accept(Poster), The reviews recognized the importance of contributions by the paper. The rebuttal by authors addressed a number of concerns, it would be helpful if the authors can address the pending concerns in the camera ready version.

---

### Decision · Program_Chairs · 2022-09-16

Accept